# Mycobacterial Cell Wall: A Source of Successful Targets for Old and New Drugs

**Catherine Vilchèze**

Einstein College of Medicine, 1301 Morris Park Avenue, the Bronx, New York, NY 10461, USA;
catherine.vilcheze@einsteinmed.org

**Abstract:** Eighty years after the introduction of the first antituberculosis (TB) drug, the treatment of drug-susceptible TB remains very cumbersome, requiring the use of four drugs (isoniazid, rifampicin, ethambutol and pyrazinamide) for two months followed by four months on isoniazid and rifampicin. Two of the drugs used in this "short"-course, six-month chemotherapy, isoniazid and ethambutol, target the mycobacterial cell wall. Disruption of the cell wall structure can enhance the entry of other TB drugs, resulting in a more potent chemotherapy. More importantly, inhibition of cell wall components can lead to mycobacterial cell death. The complexity of the mycobacterial cell wall offers numerous opportunities to develop drugs to eradicate *Mycobacterium tuberculosis*, the causative agent of TB. In the past 20 years, researchers from industrial and academic laboratories have tested new molecules to find the best candidates that will change the face of TB treatment: drugs that will shorten TB treatment and be efficacious against active and latent, as well as drug-resistant TB. Two of these new TB drugs block components of the mycobacterial cell wall and have reached phase 3 clinical trial. This article reviews TB drugs targeting the mycobacterial cell wall in use clinically and those in clinical development.

**Keywords:** tuberculosis; discovery; mode of action; drug resistance; toxicity; target

## 1. Introduction

In 1882, when Robert Koch made his ground-breaking announcement that he had discovered, isolated and cultured the bacterium responsible for tuberculosis (TB), there were no curative options for people infected with TB. In Roman times, the personal physician of the emperor Marcus Aurelius (161–180) was prescribing sea trips, fresh air and milk to treat TB patients [1]. English, French and German physicians revisited this concept 17 centuries later. In 1840, the English physician George Bodington wrote an essay "On the treatment and cure of pulmonary consumption," where he recommended fresh, cold and open air, exercise, a healthy diet and wine to treat TB patients. He opposed the treatment popular at the time: confinement of TB patients and the use of drugs such as digitalis and antimony potassium tartrate [2]. The journal Lancet published a harsh review of this essay, describing it as "very crude ideas and unsupported assertions" [3], but later admitted that they had been wrong [4]. Other English physicians followed Bodington's ideas of fresh-air treatment for TB patients but were dismissed by the medical intelligentsia [5]. On the other side of the Channel, the French physician Amédée Latour prescribed sunshine, fresh air, exercise and rich food containing 1/8 oz of sea salt every morning to treat TB [6]. The German physician Hermann Brehmer advocated high altitude, fresh air, exercise and a rich diet with some alcohol for the treatment of TB patients [5]. He opened the first sanatorium in 1854 in Görbersdorf, Prussia, to implement his concept. Sanatoria were beneficial to TB patients with early stages of disease. Lower mortality rates were observed in sanatoria compared to TB patients treated at home [1,7]. Sanatoria would close one hundred years later with the introduction of the first multiantimycobacterial drug regimen to treat TB.

The first drug against the etiologic agent of TB, ***Mycobacterium tuberculosis,*** was streptomycin, isolated from the soil bacterium ***Streptomyces griseus*** and shown to have activity against *M. tuberculosis* in 1944 [8,9]. Streptomycin was tested on a 21-year-old woman with advanced pulmonary TB and gave "impressive therapeutic effects" [10,11]. Unfortunately, resistance to streptomycin developed quickly [12,13]. In 1946, Jorgen Lehmann published the discovery of the antimycobacterial activity of para-aminosalicylic acid (PAS) in vitro, in guinea pigs and in TB patients [14]. The addition of PAS to streptomycin treatment drastically reduced the emergence of streptomycin-resistant strains in TB patients but did not abolish it [15]. It would take the introduction of a new TB drug, isoniazid (INH), in 1952 to achieve a successful treatment for TB [16].

INH is one of the most effective drugs against *M. tuberculosis*, which is still used to this day to treat active and latent *M. tuberculosis* infections. INH is part of a four-drug regimen (INH, rifampicin (RIF), ethambutol (EMB) and pyrazinamide (PZA)) to treat drug-susceptible *M. tuberculosis* infection. While RIF targets the RNA polymerase RpoB and PZA's mechanism of action is still unclear, INH and EMB target the mycobacterial cell wall.

The *M. tuberculosis* cell wall has an intricate and unique structure composed of a thick peptidoglycan layer and an outer membrane made up mostly of various lipopolysaccharides and fatty acids with imbedded glycolipids and wax esters. This lipid-rich cell wall forms a low-permeability barrier that protects *M. tuberculosis* against most antibiotics. This is one of the many challenges facing TB drug development and the main reason drug target-based screening has been rather unsuccessful [17,18]. Bacteria require an intact cell membrane in order to survive; therefore, the biosynthesis of cell wall components could be considered a weakness to exploit with new and more potent drugs. The ever-expanding threat of drug resistance should motivate greater alacrity in developing these new therapies. Global drug surveillance data from the World Health Organization indicates that in 2018, half a million people were infected with rifampicin- or multidrug-resistant (MDR) TB, and 6% of them had extensively drug-resistant (XDR) TB [19]. While an MDR *M. tuberculosis* strain is "only" resistant to INH and RIF, an XDR strain is resistant to at least five TB drugs (INH, RIF and three second-line TB drugs). Treatment for drug-resistant TB is very long (minimum 20 months), requiring multiple drugs with potential severe adverse reactions. MDR TB treatment's success rate is approximately 55%, whereas XDR TB is barely 39%. New drugs and shorter drug regimens are actively sought to increase these success rates. This article covers TB drugs specifically targeting the mycobacterial cell wall currently used in clinics or in clinical development, focusing on their discovery, activity, toxicity, mode of action and resistance.

## 2. The Mycobacterial Cell Wall

The biosynthesis of the mycobacterial cell wall components has been extensively described recently [20–22], and only a summary of the most pertinent points for this article is presented here. The proteins implicated or targeted by TB drugs discussed below are underlined.

The three main components of the cell wall are the peptidoglycan (PG), the lipopolysaccharides (arabinogalactan, lipoarabinomannan, and lipomannan) and the outer membrane, which contains mycolic acids, various glycolipids and phthiocerol dimycocerosates (Figure 1).

The *M. tuberculosis* peptidoglycan is located outside of the mycobacterial inner membrane, conferring rigidity, integrity and shape to the cell [22]. The peptidoglycan is a polysaccharide composed of alternating *N*-acetylglucosamine and muramic acid (either *N*-acetylated or *N*-glycolylated) residues linked by β (1→4) bonds [23–26]. Strands of polysaccharides are acylated on the muramic acid residues by the pentapeptide L-alanyl-D-isoglutaminyl-*meso*-diaminopimelyl-D-alanyl-D-alanine synthesized by the Mur ligases (MurC/D/E/F). The acylation reaction is performed by the D-Ala:D-Ala ligase DdlA. The pentapeptides are cross-linked to form the peptidoglycan. Two types of cross-linkages are observed in *M. tuberculosis*: (i) the D,D-transpeptidase activity of the penicillin binding proteins PonA1 and PonA2 cross-links meso-diaminopimelic acid and D-alanine to form a 3→4 linkage; (ii) the

L,D-transpeptidases (LdtMt1 to 5) cross-link two meso-diaminopimelate residues to form a 3→3 linkage [22].

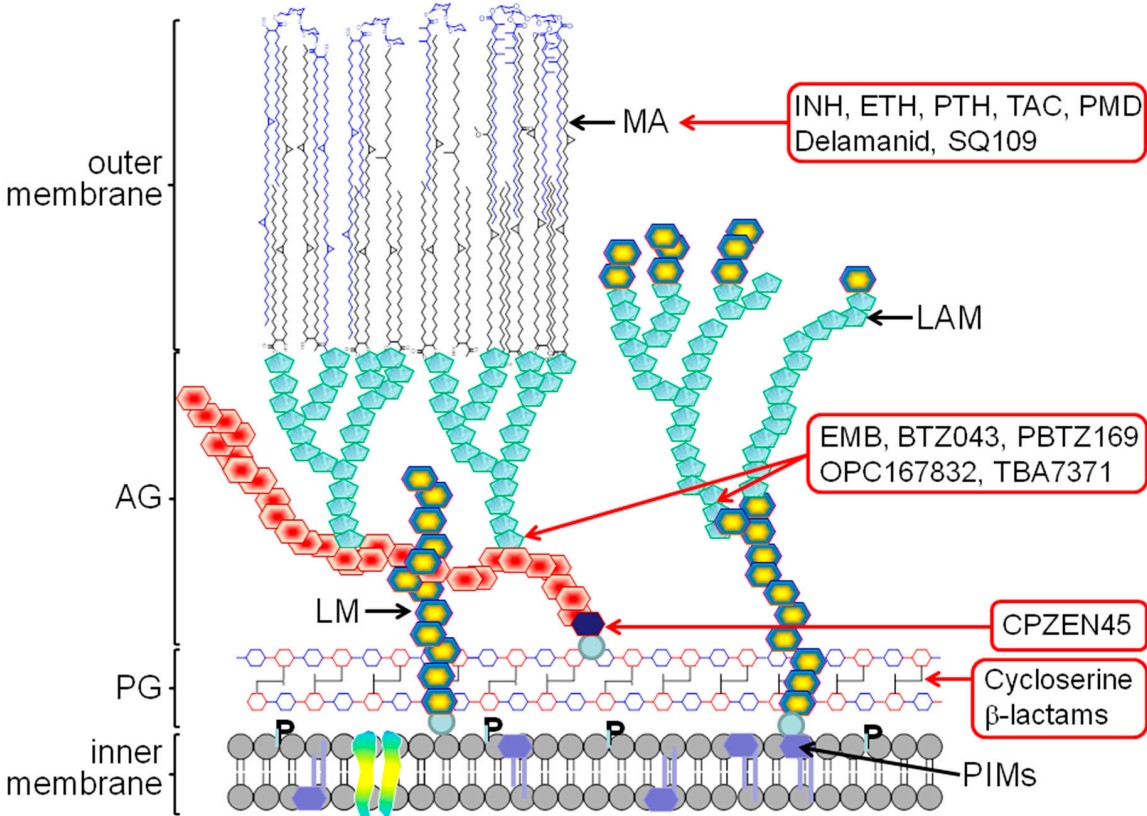

**Figure 1.** Illustration of the mycobacterial cell wall. The outermost layer of the cell wall, the capsule, is omitted. The cell wall components/sites inhibited by the tuberculosis (TB) drugs discussed herein are indicated. Abbreviations: AG, arabinogalactan; LAM, lipoarabinomannan; LM, lipomannan; MA, mycolic acids; PG, peptidoglycan; P, phospholipid; PIMs, phosphatidyl-*myo*-inositol mannosides; INH, isoniazid; ETH, ethionamide; PTH, prothionamide; TAC, thiacetazone; PMD, pretomanid; EMB, ethambutol.

The arabinogalactan is a branched polysaccharide composed of arabinose (Ara*f*) and galactose (Gal*f*) residues in the furanose configuration [27]. The first step in the arabinogalactan biosynthesis is the formation of the linker that anchors the arabinogalactan complex to the peptidoglycan via the *N*-glycolylated-muramic acid residues [28]. This linker is a decaprenyl-diphospho-*N*-acetylglucosamine-rhamnosyl molecule produced by the successive transfer of *N*-acetylglucosamine-1-phosphate to decaprenyl-phosphate by WecA followed by the transfer of L-rhamnose by WbbL. On this linker, 30 linear Gal*f* residues are added by the galactofuranosyl transferases GlfT1 and GlfT2. An Ara*f* unit is then transferred to the galactan chain using the arabinose donor decaprenylphosphoryl-D-arabinose (DPA). DPA is formed through several steps, starting with phospho-α-D-ribosyl-1-pyrophosphate (pRpp). UbiA adds a decaprenyl group to form decaprenol-1-monophosphate 5-phosphoribose (DPPR), which is dephosphorylated and epimerized by DprE1 and DprE2 to form DPA. The arabinofuranosyltransferase AftA catalyzes the transfer of the first unit of Ara*f* to the galactan chain. The arabinosyltransferases EmbA and EmbB catalyze the further addition of Ara*f* to form the arabinan. The final product, the arabinogalactan, is a linear galactan to which highly branched arabinans are attached. The arabinan anchors the mycolic acids forming the mycolyl-arabinogalactan-peptidoglycan (mAGP) complex.

Mycolic acids are the hallmark of the *M. tuberculosis* cell wall, an essential component regulating the permeability, acid-fast staining, viability and virulence of *M. tuberculosis* [29]. As stated above,

mycolic acids are found attached to the arabinose part of the arabinogalactan complex but can also be found as a free form or bound to other saccharides to form trehalose mono/dimycolates (TMM/TDM) and glucose monomycolate. Mycolic acids are long-chain fatty acids composed of a meromycolate chain containing up to 62 carbons with various modifications (*cis/trans* cyclopropane, keto, or methoxy groups) to which a saturated $C_{26}$ alkyl chain is attached at the α position (Figure 2). The biosynthesis of mycolic acids starts with two fatty acid synthases (FAS): the eukaryotic-like type I (FAS-I) and the prokaryotic-like type II (FAS-II) (Figure 2). FAS-I is a multidomain polypeptide [30] that synthesizes the α-$C_{26}$ alkyl chain and also provides a $C_{14/16}$ fatty acyl-CoA to be elongated into the meromycolate chain by the FAS-II system. This elongation reaction starts with the condensation of $C_{14/16}$ fatty acyl-CoA with malonyl-Acyl Carrier Protein (ACP) by the β-ketoacyl-ACP synthase III FabH. The resulting β-ketoacyl-ACP intermediate is delivered to FAS-II, which performs cycles of elongation using independent enzymes: the reductase MabA, the heterodimer dehydratases HadAB and HadBC, the enoyl-reductase InhA and the condensases KasA and KasB (Figure 2). The modifications (*cis/trans* cyclopropane, keto, or methoxy groups) of the meromycolate chains are introduced either during the elongation by FAS-II or when the meromycolate chain is fully formed [31]. The meromycolate chain is activated by the fatty acid AMP synthase FadD32 to a meromycolyl-AMP, which is coupled to the carboxylated α-$C_{26}$ fatty acyl-CoA (from FAS-I) by the polyketide synthase Pks13. Reduction of the resulting mycolic β-ketoester by the mycolyl reductase CmrA yields a mature mycolic acid. The biosynthesis of mycolic acids takes place in the cytoplasm. Transfer of the mycolic acids to the cell envelope occurs via the formation of a trehalose monomycolate (TMM) which is translocated by the efflux pump MmpL3 (Mycobacterial membrane protein Large). The mycolyltransferase Antigen 85 complex (*fbpA, fbpB, fbpC*) then condenses mycolic acids to the arabinogalactan releasing a molecule of trehalose.

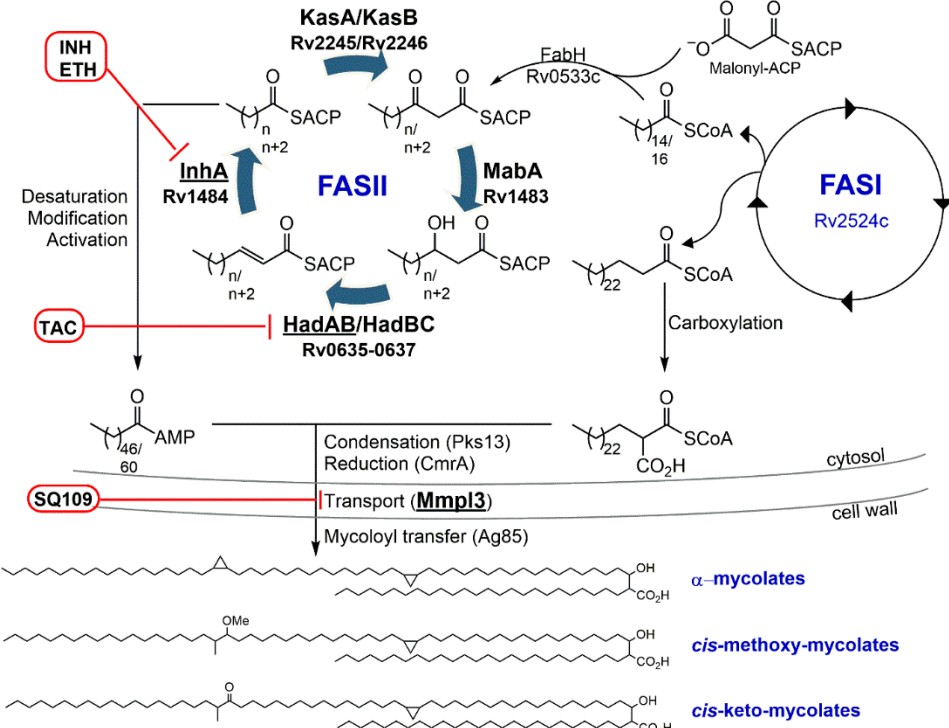

**Figure 2.** Schematic representation of the biosynthesis of mycolic acids. The three families of mycolic acids in *Mycobacterium tuberculosis* are the α-, methoxy- and keto-mycolic acids. Only the cis configuration of the cyclopropane group of the methoxy- and keto-mycolic acids is shown, but *M. tuberculosis* also produces the trans-methoxy- and trans-keto-mycolic acids. The known targets of the mycolic acid inhibitors in clinical use or in clinical trial are InhA for isoniazid (INH) and ethionamide (ETH); HadAB for thiacetazone (TAC); and Mmpl3 for SQ109.

Besides the mAGP complex, the mycobacterial cell wall also includes lipomannan (LM), lipoarabinomannan (LAM) and phosphatidyl-*myo*-inositol mannosides (PIMs). These lipids are anchored in the inner membrane and play an important role in *M. tuberculosis* growth, survival and virulence [32–34]. In the outer membrane, trehalose-containing glycolipids and phthiocerol dimycoserates are found interlaced with the mycolic acids. These lipids are crucial for their interactions with the host and its immune system [35].

Specific components of the cell wall are essential for *M. tuberculosis* survival in the host. It is, therefore, not surprising that two of the four first-line TB drugs target the cell wall. In the past 20 years, potent new TB drugs have been discovered that target the peptidoglycan, arabinogalactan, LAM, and mycolic acids or their transfer to the cell wall. This article focuses on the cell wall inhibitors included in the TB pharmacopeia since the 1950s/1960s (old TB drugs) and the new ones in clinical development.

## 3. Old TB Drugs

### 3.1. Isoniazid—Target: Mycolic Acids

***Discovery.*** In 1949, Colin Hinshaw and Walsh McDermott went to Germany to investigate reports that TB patients were successfully treated with a new synthetic compound, Conteben [36]. Conteben, a thiosemicarbazone also known as thiacetazone and Tibione (see thiacetazone section below), had been discovered by Gerhard Domagk at Bayer in West Germany. Hinshaw and McDermott returned to the USA with a supply of Conteben. After testing of Conteben in US hospitals, they concluded that "a prompt and thorough series of experimental and clinical trials in the United States" was justified along with experimenting with other thiosemicarbazones [37]. Two US pharmaceutical companies, Hoffman-La Roche and E. R. Squibb & Sons, quickly developed a series of thiosemicarbazone analogs but none showed better activity than Conteben [38–40] until the benzene ring in Conteben was replaced with a pyridine ring leading to the simultaneous discovery of Rimifon at Hoffman-La Roche [39] and Nydrazid at Squibb [41]. In parallel, Domagk at Bayer developed the thiosemicarbazone analog Neoteben [42]. Rimifon, Nydrazid and Neoteben had antiTB activity that far exceeded streptomycin, PAS and any other analog synthesized so far and share the same chemical structure: 4-pyridinecarboxylic acid hydrazide. Hoffman-La Roche, Squibb and Bayer had simultaneously and independently discovered the most potent TB drug at the time: isoniazid (INH). Ironically, none of the pharmaceutical companies could patent their discovery, as INH had been synthesized 40 years earlier by two Polish graduate students Hans Meyer and Josef Mally [36]. The results with INH were so striking that chemotherapy became the leading route for TB treatment, and soon after the sanatoria closed. Almost 70 years later, INH remains the cornerstone of TB chemotherapy for the treatment of drug-susceptible and latent *M. tuberculosis* infections.

***Activity and toxicity.*** INH is a first-line TB drug. INH is an oral, highly soluble in water (140 g/L), bactericidal drug with a minimum inhibitory concentration (MIC) ranging from 0.1 to 0.7 μm against *M. tuberculosis*. In vitro, INH rapidly reduces the number of *M. tuberculosis* bacteria by 2- to 3-$\log_{10}$s during the first four days of treatment [43]. This bactericidal activity is only observed in exponentially growing *M. tuberculosis* cultures. INH has no activity in stationary or persistent *M. tuberculosis*. In mice, a similar pattern is observed, where INH is bactericidal only on actively dividing *M. tuberculosis* [44]. INH is readily absorbed and reaches concentrations in tissues and organs above its MIC.

The main adverse effect of INH is hepatotoxicity. TB patients more likely to develop liver damage when taking INH are slow acetylators [45,46]. INH is acetylated into the inactive molecule AcINH by the human *N*-acetyltransferase-2 (NAT2), which is expressed mostly in the liver and gastrointestinal tract. Genetic polymorphisms in NAT2 renders this acetylation reaction either slow or fast, dividing TB patients between slow and fast acetylators. INH metabolism also involves an amidase metabolizing INH into an hydrazine (Hz), which is then acetylated by NAT2 to form a toxic acylhydrazine (AcHz) and a non-toxic diacylhydrazine (DiAcHz) [46]. Hz, AcHz and their metabolites generated by the

liver microsomal cytochrome P450 enzymes have been linked to liver toxicity [47,48]. Fast acetylators have a lower risk of liver toxicity by producing more of the DiAcHz metabolite and less of the AcHz metabolite(s) [49].

*Mode of action.* INH enters *M. tuberculosis* cells by passive diffusion as a prodrug. A prodrug is the ideal drug where the compound has, if possible, no effect on eukaryotic cells, yet, once activated by a pathogen-specific enzyme, leads to the death of the pathogen [50]. Curiously, the TB pharmacopeia is composed of an array of prodrugs approved for clinical therapy as well as in the developmental phase [51]. INH is activated by the mycobacterial catalase peroxidase KatG (Rv1908c) [52] into what is most likely an isonicotinoyl radical that reacts with nicotinamide adenine dinucleotide (NAD$^+$) to form the INH-NAD adduct [53–56]. This adduct binds to and inhibits InhA [54,57–60], the NADH-dependent enoyl-ACP reductase [61,62] of the FAS-II system [63], leading to the inhibition of mycolic acid biosynthesis and mycobacterial cell death [60,64–66]. Although the mechanism of INH action seems rather straightforward, elucidation of the molecular details took almost fifty years [67].

*Resistance.* Shortly after the antimycobacterial activity of INH was published in 1952, the first report of INH-resistant clinical isolates appeared less than one year later [68]. In 1954, Gardner Middlebrook demonstrated that INH-resistant mutants isolated in vitro were catalase-negative [69]. It will then take another 40 years to discover the genetic basis of this phenotype and decipher the main mechanism of resistance to INH: mutation in *katG,* the gene encoding the INH activator. Zhang and colleagues showed that 1) complementation of an INH-resistant *Mycobacterium smegmatis* mutant with a single copy of *katG* restored INH susceptibility [52]; 2) two highly INH-resistant clinical isolates had *katG* deletion [52]; and 3) *M. tuberculosis* INH-resistant mutants regained INH susceptibility when transformed with *katG* [70]. Since then, more than 300 *katG* mutations covering 99% of the gene's length (*katG* has 2223 base pairs (bp)) have been identified in INH-resistant laboratory and clinical strains [71]. The most frequent mutation in clinical isolates is the Ser315Thr. Actually, each of the three bases of the serine codon (AGC) can be mutated leading to Asn, Arg, Ile, Gly or Leu amino acid change. Mutations in KatG alter its catalase peroxidase and oxidase activities causing a defect in KatG's ability to activate INH [72–74]. Thus, most *M. tuberculosis* clinical strains carrying *katG* mutations are highly resistant to INH [75]. KatG enzymatic activities can also be disrupted by mutations in *furA* (*Rv1909c*), a gene encoding a ferric uptake regulation protein and a negative regulator of *katG* transcription. Isogenic strains carrying the mutations a-10c and g-7a in the intergenic region between *katG* and *furA* had reduced *katG* expression leading to a decrease in INH oxidase activity and a modest increase in INH resistance [76].

Resistance to a drug can occur through either target mutation (preventing the binding of the drug to its target) or target overexpression (titration of the drug). This is the case for the second most common mutations in INH-resistant clinical isolates: mutations in *inhA* and its promoter region. The c-15t mutation in the *inhA* promoter region increases *inhA* mRNA levels by 20-fold resulting in higher InhA protein levels and an 8-fold MIC increase in *M. tuberculosis* [60]. This mutation is found in about one third of the INH-resistant clinical isolates but more often in XDR TB cases than MDR or INH-monoresistant TB cases, suggesting that the c-15t mutation could be a marker for XDR TB [77]. In contrast to *katG* mutations, clinical strains with *inhA* mutations (either in the gene or in the promoter region) have a low INH resistance phenotype [75]. There are another 20 different mutations identified in the *inhA* promoter region [71]. Mutations in the target of INH *inhA*, an essential gene, are rare, with only 17 identified so far [71]. The first *inhA* mutation (Ser94Ala) was isolated in vitro in *M. smegmatis* during a screening for mutants co-resistant to INH and ethionamide (ETH), a second-line TB drug [57]. This mutant led to the hypothesis that InhA was the primary target of both INH and ETH [57]. Introduction of the Ser94Ala mutation into wild-type *M. tuberculosis* H37Rv was shown to be sufficient to confer INH and ETH resistance [60] and to decrease the binding of the INH-NAD adduct to InhA [60,78]. These observations strongly supported the conclusion that INH and ETH target InhA. The Ser94Ala mutation has been found in INH-resistant *M. tuberculosis* clinical isolates carrying no mutations in *katG* [75,79–81].

Mutations in many other genes have been identified in INH-resistant laboratory and clinical strains such as *kasA, mshA, ndh, nudC, ahpC,* and *nat* (the *M. tuberculosis N*-acetyltransferase) to cite a few, but very often these mutations were present in clinical strains already carrying a *katG* or *inhA* mutation or were also found in INH-susceptible strains questioning their roles in INH resistance [71].

*Area of investigation.* For the past 70 years, INH has had an essential role in TB treatment and control. One of INH downsides is its lack of activity against dormant/persistent *M. tuberculosis*. The reasons for this lack of activity in the dormant form of *M. tuberculosis* are still up for debate; however, it is known that KatG has limited activity in the dormant state [44]. With KatG being a major factor in INH resistance and a potential player in INH shortcoming in dormant *M. tuberculosis*, new InhA inhibitors that would not require activation by KatG have been actively sought [82–89]. Compounds such as GlaxoSmithKline's thiadiazole GSK693 [90] or the diazaborine AN12855 [91,92] are promising leads with good oral bioavailability, low toxicity, activity against *katG*-deficient *M. tuberculosis* and in vivo efficacy similar to INH.

### 3.2. Ethambutol—Target: Arabinogalactan/LAM

*Discovery.* Ethambutol ((+)-2,2′-(ethylenediimino)di-1-butanol, EMB) was discovered at the Lederle Laboratories division of the American Cyanamid Company in New York (USA) in 1961. During a random screening of synthetic compounds in mice, *N,N′*-diisopropylethylenediamine was found to protect mice from an *M. tuberculosis* infection. Following an intensive campaign of structure–activity relationship (SAR), ethambutol, a di-hydroxylated derivative of *N,N′*-diisopropylethylenediamine, was synthesized [93–95]. Its activity against *M. tuberculosis* in infected mice was four times more potent than streptomycin and protected mice infected with streptomycin- or isoniazid-resistant *M. tuberculosis* strains.

*Activity and toxicity.* EMB is water soluble (solubility 7.58 g/L) and easily taken up by *M. tuberculosis* [96]. EMB is bacteriostatic, with a MIC ranging from 5 to 34 μm against *M. tuberculosis*. EMB is a first-line TB drug, given for the first two months of TB treatment alongside INH, RIF and PZA. EMB adverse effects include liver and ocular toxicity (decreased vision, color blindness) although these effects are reversible once EMB treatment is stopped [97].

*Mode of action.* The target of EMB has not been definitively determined. It was initially thought that EMB inhibited the synthesis of metabolites needed for *M. tuberculosis* replication [96] or hampered RNA synthesis [98]. A set of studies in *M. smegmatis* determined that EMB treatment resulted in the inhibition of mycolic acid transfer to the cell wall [99]; the cellular accumulation of TMM, TDM and free mycolic acids [100]; the inhibition of D-arabinose incorporation into the arabinogalactan complex and arabinomannan [101]; and the accumulation of DPA [102], the arabinose donor in the synthesis of arabinan [103]. While the synthesis of arabinan for the arabinogalactan complex was completely inhibited by EMB, EMB only partially inhibited the synthesis of the arabinan of LAM [104], leading to the conclusion that EMB inhibited different arabinosyl transferases responsible for the formation of arabinan in the cell wall [105].

Recently, EMB was postulated to target the glutamate racemase, MurI (Rv1338) [106]. MurI racemizes L-glutamate to D-glutamate, which is required for peptidoglycan biosynthesis. In an enzymatic assay, EMB partially inhibited the MurI racemisation reaction. Docking experiments suggested that EMB could bind to MurI and act as a competitive inhibitor of MurI substrate. Further experiments are required to confirm the role of MurI in *M. tuberculosis* inhibition by EMB.

*Resistance.* Characterization of EMB-resistant *M. tuberculosis* clinical isolates revealed a connection between EMB resistance and a cluster of genes (*embCAB, Rv3793–3795*) [107,108] encoding arabinosyltransferases involved in the biosynthesis of the mycobacterial cell wall arabinan [109]. Initially, the most common mutations found in EMB-resistant but not in EMB-susceptible clinical isolates were at position 306 of *embB* (M306I, M306V and M306L) [107]. When the *embB* mutations M306L, M306V and M306I were introduced into wild-type *M. tuberculosis* strains, the resulting strains

were found to be EMB-resistant, suggesting that the *embB* mutations were a molecular marker for EMB resistance [110,111]. However, *embB* mutations were found in EMB-susceptible clinical isolates, leading to questioning the role of *embB* in EMB resistance [112–115]. Additionally, EMB-monoresistant clinical isolates did not harbor mutation in *embB*, suggesting a different target for EMB [113]. EMB-susceptible strains carrying *embB* mutations were resistant to other antituberculosis drugs pointing to the *emb* locus as a marker for MDR or XDR *M. tuberculosis* strains [110,112,115].

Safi and colleagues have postulated that the high-level EMB resistance found in clinical isolates was a result of an accumulation of various mutations, each individually causing a low-level resistance to EMB but together provided a high degree of resistance [110]. The authors demonstrated that, in vitro, a stepwise introduction of specific mutations in *embB, Rv3806c* (*ubiA*) and *embC* in *M. tuberculosis* led to a strain 8-fold more resistant to EMB than wild-type *M. tuberculosis*. *ubiA* encodes a DPPR synthase involved in DPA biosynthesis. Mutations in *ubiA* increase intracellular DPA levels, which might bind to Emb, leading to EMB resistance [116]. However, the role of *ubiA* in EMB resistance is questionable since mutations in *ubiA* are found in both EMB-susceptible and EMB-resistant *M. tuberculosis* clinical isolates [117].

*Area of investigation.* SAR on ethambutol was performed to improve EMB antimycobacterial activity while reducing its toxicity. The outcome was SQ109, a compound now in phase 2 clinical trial, and with a very different mechanism of action than EMB (see below).

### 3.3. Ethionamide/Prothionamide—Target: Mycolic Acids

*Discovery.* In 1954, Thomas Gardner and colleagues published the synthesis of an INH analog, where a thioamide group replaced the acyl hydrazide group in INH [118]. This thioisonicotinamide compound was active in a mouse model of *M. tuberculosis* infection but was found to be less potent than INH. This publication was noticed by a French team who found that the thioamide derivative was more active than streptomycin and more importantly, potent against INH-, streptomycin- and PAS-resistant *M. tuberculosis* strains. They synthesized a series of α-alkyl derivatives of the thioisonicotinamide and found that adding an ethyl or propyl group at the α position generated two compounds with activity in vitro and in mice greater than streptomycin but not as potent as INH [119]. They had synthesized ethionamide (ETH) and prothionamide (PTH).

*Activity and toxicity.* ETH is a bactericidal drug, with a MIC ranging from 6 to 15 μm against drug-susceptible *M. tuberculosis.* ETH and PTH are poorly soluble in water (0.84 and 0.28 g/L, respectively). ETH and PTH are in the group C (other core second-line agents) of second-line TB drugs used interchangeably for the treatment of MDR and XDR TB cases. ETH and PTH are oral drugs with some severe adverse effects (hepatoxicity, gastrointestinal disorders, neurotoxicity) [120].

*Mode of action.* Winder and colleagues were the first to demonstrate that ETH inhibited mycolic acid biosynthesis [121]. They noticed that ETH affected *Mycobacterium bovis* BCG similarly to INH although they observed no cross resistance between INH and ETH leading to the conclusion that the mode of action of INH and ETH were not identical [121]. Actually, cross resistance between INH and ETH had been observed in TB patients a few years earlier. Several studies noted that TB patients treated with INH developed resistance to both INH and ETH although the patients had never received ETH [122–124]. This suggested that ETH and INH shared a common mechanism of action. Winder and colleagues had postulated that INH and ETH "might differ in the means by which they reach the sensitive site". That was prescient since INH and ETH were eventually revealed as prodrugs activated by different enzymes. While INH is activated by a catalase peroxidase, ETH is activated by the NADPH-specific flavin adenine dinucleotide-containing Baeyer–Villiger monooxygenase EthA (Rv3854c also called EtaA) [125,126]. In vitro, activation of ETH by EthA leads first to the formation of ethionamide S-oxide, which is further metabolized by EthA to form either 2-ethyl-4-amidopyridine [126] or (2-ethyl-pyridin-4-yl) methanol [125] via radical intermediate(s). Once activated, the mechanism of action of ETH is very similar to INH. The activated form of ETH

reacts with $NAD^+$ to form an ETH*-NAD adduct [127]. The structures of this adduct as well as the PTH*-NAD adduct were determined by X-ray crystallography, revealing an ethyl-isonicotinoyl or a propyl-isonicotinoyl covalently attached to the nicotinamide portion of reduced $NAD^+$ and bound to InhA. Inhibition of InhA by these adducts results in mycolic acid biosynthesis inhibition.

Wang and colleagues postulated that the activated form of ETH was an iminoyl radical but expressed doubt that this activation reaction was caused by EthA alone and hypothesized that additional enzymes might be involved [127]. Another study also concluded that another mechanism of activation might exist for ETH when demonstrating that deletion of *ethA* and its regulator *ethR* (*Rv3855*) in *M. tuberculosis* caused only a modest increase (3-fold) in ETH resistance [128]. EthA has two close homologs: the monooxygenases Rv3083 (MymA) and Rv0565c [125]. Grant and colleagues determined that MymA was indeed an activator of ETH by showing that loss of function of MymA or overexpression of *mymA* conferred resistance or hypersusceptibility to ETH, respectively [129]. In this study, the authors also noted that transposon mutants in *Rv0565c* as well as in two other genes encoding Baeyer–Villiger monooxygenases (*Rv1393c* and *Rv3049c*) did not lead to ETH resistance. However, a recent study determined that overexpression or deletion of *Rv0565c* led to ETH hypersusceptibility or low resistance, respectively, in *M. tuberculosis*, leading the authors to conclude that Rv0565c was an additional activator of ETH in *M. tuberculosis* [130]. In both studies on *mymA* and *Rv0565c*, no biochemical evidence was provided to show that MymA or Rv0565c actually activates ETH and what would be the resulting activated molecule(s).

<u>*Resistance.*</u> Mutations in ETH-resistant laboratory and clinical *M. tuberculosis* strains are found in the activators of ETH (*ethA, mymA, Rv0565c*), the negative regulator of *ethA* (*ethR*) and ETH target (*inhA* gene and promoter region) [129–131]. Mutations in the *inhA* promoter region are more frequent than mutations in the activator(s) of ETH [131]. Mutations in *inhA* or in its promoter region are found in up to 68% of ETH-resistant clinical isolates while mutations in ETH activator *ethA* are usually found in no more than 55% of ETH-resistant clinical isolates, and some of these *ethA* mutations can also be found in ETH-susceptible strains [131]. The mutations in *ethA* cover 91% of the gene (from 2 to 1341 bp; *ethA* has 1470 bp) [71]. Mutations in *ethR* have been identified in ETH-resistant clinical strains carrying the c-15t mutation in the *inhA* promoter region [131]. Recently, cyclic dimeric guanosine monophosphate, a bacterial second messenger, was shown to boost the binding of EthR to *ethA* promoter, causing a decrease in *ethA* transcription levels and ETH resistance [132].

Other genes have been implicated in ETH resistance. *M. tuberculosis* strains deleted for *mshA*, a gene encoding the glycosyltransferase involved in the biosynthesis of mycothiol, a major low-molecular-weight thiol, are eight times more resistant to ETH than their parental strains [133]. Mycothiol is thought to play a role in ETH resistance by increasing the rate of ETH activation by EthA [133]. In *M. smegmatis* and *M. bovis* BCG, mutants co-resistant to INH and ETH were isolated carrying mutations in *ndh*, a gene encoding an NADH dehydrogenase whose function is to oxidize NADH into $NAD^+$. In this case, *ndh* mutants were shown to accumulate NADH. Excess NADH could act as a competitive inhibitor for the binding of the ETH-NAD to InhA, triggering ETH resistance [134].

<u>***Area of investigation.***</u> Regulation of *ethR* expression plays a role in ETH resistance and susceptibility. While overexpression of *ethR* was shown to cause ETH resistance, inhibition of *ethR* triggers higher levels of *ethA* expression and increases ETH susceptibility [135]. Baulard and colleagues have taken advantage of this point and generated EthR inhibitors to boost ETH activity. Screening of chemical libraries and SAR on potential EthR inhibitors led to a first series of EthR inhibitors, which by themselves had no activity on *M. tuberculosis*, yet, when combined with ETH, significantly "boosted" ETH activity in vitro and in *M. tuberculosis*-infected mice [136]. The most active of this first generation of boosters, the oxadiazole BDM41906, decreased the MIC for ETH to the nM range [137]; however, BDM41906 had no "boosting" activity in *M. tuberculosis* strains carrying *ethA* mutations [138]. The second generation of ETH booster, the spiroisoxazoline SMARt (Small Molecule Aborting Resistance)-420, does not inhibit EthR but instead triggers a different activation mechanism for ETH. This new activation system uses the oxidoreductase Rv0077c as the activator,

which is negatively regulated by Rv0078. SMARt-420 binds to Rv0078, releasing the activity of Rv0077c. SMARt-420 increases the susceptibility of *M. tuberculosis* to ETH 40-fold. In addition, ETH-resistant *M. tuberculosis* strains carrying an *ethA* mutation regain ETH susceptibility when SMARt-420 is present, both in vitro and in mice [138]. SMARt-420 can, therefore, be used in the context of ETH resistance due to *ethA* mutations but not due to *inhA* mutations.

### 3.4. Cycloserine—Target: Peptidoglycan

*Discovery.* In contrast to INH, EMB and ETH, which are synthetic compounds, cycloserine is a natural product produced by *Streptomyces* with broad-spectrum activity. Cycloserine (D-4-amino-3-isoxazolidone, also called oxamycin or seromycin) was isolated from *Streptomyces garyphalus* [139], *Streptomyces lavendulae* [140], *Streptomyces roseochromogenus* [141] and *Streptomyces orchidaceus* [142] in the early 1950s.

*Activity and toxicity.* Cycloserine is an oral drug, highly soluble in water (877 g/L). Cycloserine is a bacteriostatic second-line TB drug, with a MIC ranging from 50 to 250 μm against drug-susceptible *M. tuberculosis*. An analog of cycloserine, terizidone, a compound made of two molecules of cycloserine, is also used as a second-line TB drug to treat MDR TB. Cycloserine was shown to inhibit the growth of *M. tuberculosis* in vitro [143]. In vivo, cycloserine is rather inactive in a mouse or guinea pig model of *M. tuberculosis* infection [143–145] but effective in *M. tuberculosis*-infected monkeys and humans [146–148]. The differences in cycloserine activity in animal models were correlated with levels of D-alanine in sera [149] and rate of cycloserine excretion [146]. In a cohort study in China, TB patients with MDR TB had a better outcome when cycloserine was added to the treatment but that was not the case for TB patients infected with pre-XDR or XDR *M. tuberculosis* strains [150]. As an inhibitor of peptidoglycan biosynthesis, cycloserine has the distinctive feature of having a unique mechanism of action thus preventing any cross resistance with other first-line and second-line TB drugs. Cycloserine is, therefore, a useful addition to second-line TB drugs, although its use might be limited by its severe toxicity. Serious adverse effects were observed during cycloserine treatment such as neuropathy and behavioral changes [151]. Cycloserine is, therefore, contraindicated in patients suffering from severe depression, suicidal tendencies, kidney failure, epilepsy or seizures.

*Mode of action.* The mechanism of action of cycloserine was primarily deciphered in *Staphylococcus aureus.* Cycloserine is an analog of D-alanine and works as an antagonist of D-alanine [152]. Cycloserine inhibits the alanine racemase Alr (Rv3423c), which converts L-alanine to D-alanine, and the D-Ala:D-Ala ligase DdlA (Rv2981c) [153] preventing the integration of alanine into the pentapeptide core of the peptidoglycan. In *M. tuberculosis*, DdlA is thought to be the primary target of cycloserine [154–156].

*Resistance.* D-cycloserine-resistant *M. tuberculosis* mutants were isolated as early as 1957 in TB patients treated with cycloserine [157]. Cycloserine resistance has been associated with mutations in the genes encoding the alanine transporter CycA (Rv1704c), the L-alanine dehydrogenase Ald (Rv2780) and the alanine racemase Alr. A Gly122Ser mutation in *cycA* is present in the naturally cycloserine-resistant *M. bovis* BCG vaccine strain [158]. Complementation of BCG with a cosmid containing *M. tuberculosis cycA* renders BCG more susceptible to cycloserine than the parental strain leading the authors to conclude that *cycA* may be a factor in cycloserine resistance in BCG [158]. Desjardins and colleagues demonstrated that deletion of *ald* increased the resistance to cycloserine 2-fold in *M. tuberculosis*, while complementation of the *ald* mutant with a plasmid expressing *M. tuberculosis ald* only partially restored cycloserine susceptibility. Notably, complementation of BCG with the *ald* plasmid did not alter the strain resistance to cycloserine [159]. Mutations in *alr* (M319T, Y364D, R373L and c-8t in the promoter region) have been identified in XDR TB strains isolated from TB patients treated with cycloserine [160]. Further, the clinical isolate with the R373L mutation in *alr* also contained a deletion in *ald*.

In an in vitro experiment, 18 cycloserine-resistant mutants were obtained by culturing *M. tuberculosis* H37Rv on plates containing increasing concentrations of cycloserine (from 0.2 to

3 mM) and characterized using whole-genome sequencing [161]. Mutants were only obtained on plates containing less than 0.8 mM of cycloserine. A mutation was identified in *alr* (D344N), but no mutations in *ddlA*, *ald* or *cycA* were found. Fifteen novel mutations were identified in genes involved in various pathways however no complementation or allelic exchange experiments were performed to confirm that these novel mutations were indeed involved in cycloserine resistance.

In a recent in vitro study [162], the authors showed that spontaneous cycloserine-resistant mutants emerged at a lower frequency ($10^{-10}$–$10^{-11}$) than rifampicin- ($10^{-9}$) or isoniazid-resistant ($10^{-8}$) mutants in *M. tuberculosis*. The authors characterized 11 independent cycloserine-resistant *M. tuberculosis* mutants by whole-genome sequencing and found mutations in *alr* (gene (D322N) and promoter region) but again not in *ddlA*, *ald* or *cycA*. None of these 11 cycloserine-resistant mutants showed any cross-resistance to other first-line and second-line TB drugs. The *alr* promoter mutation upregulated *alr* gene transcript, causing an increase in Alr protein level by up to 30-fold. The *alr* D322N mutation present in 8/11 mutants reduced the binding affinity for cycloserine by 240 fold while having limited effect on Alr enzymatic activity. Evangelopoulos and colleagues postulated that the *alr* mutations protected the enzymatic function of both cycloserine targets by decreasing the affinity of cycloserine to Alr and increasing the levels of D-alanine preventing the binding of cycloserine to DdlA. Their main conclusion was that DdlA was the "lethal target" for cycloserine.

### 3.5. Thiacetazone—Target: Mycolic Acids

*Discovery*. Gerhard Domagk was awarded the Nobel Prize in 1939 for the discovery of prontosil, a prodrug that releases *p*-aminobenzenesulfonamide, the first sulfonamide drug active against *Staphylococcus* and *Streptococcus* infections, but not against *M. tuberculosis* infections [163]. While prontosil was quickly replaced by penicillin to fight antibacterial infections, Domagk and his team continued working on the synthesis of related compounds, the thiosemicarbazides, and discovered that one in particular, thiacetazone (TAC, also called Tibione, Tb I, Conteben), had impressive activity against *M. tuberculosis* in guinea pigs [164]. In the late 1940s, TAC was tested in Germany on patients with various forms of TB and found to have promising activity although severe adverse effects were recorded [37,165].

*Activity and toxicity*. TAC is an oral, inexpensive, effective and bacteriostatic drug, with a MIC ranging from 0.3 to 5 μm against drug-susceptible *M. tuberculosis*. TAC is poorly soluble in water (0.09 g/L). TAC is part of the group D3 (add-on agents) of second-line TB drugs. TAC has been given in combination with INH in resource-poor countries [166,167]. Its utilization as a TB drug was mostly discontinued because of its high toxicity (skin disorder, gastrointestinal symptoms, conjunctivitis, vertigo, liver and kidney damage) especially in HIV-positive TB patients [168]. TAC is, therefore, not recommended in HIV-positive TB patients.

*Mode of action*. TAC is a prodrug that is activated by the monooxygenase EthA [127,169]. Once activated, TAC inhibits the dehydratase HadAB of the FAS-II system by forming a disulfide bound with a cysteine (Cys61) residue of HadA [170,171]. During the elongation of fatty acyl-ACPs by FAS-II, HadA binds the acyl-ACP while HadB performs the dehydratase reaction. When covalently bound to the Cys61 residue of HadA, TAC obstructs the acyl-ACP channel preventing the binding of the fatty acyl-ACP [172]. The covalent bond between TAC and HadA Cys61 blocks the activity of HadAB leading to inhibition of mycolic acid biosynthesis [173].

*Resistance*. The mode of action of TAC was discovered through analysis of TAC resistance. Since TAC treatment of *M. tuberculosis* results in inhibition of mycolic acids, Belardinelli and Morbidoni overexpressed every gene involved in the FAS-II system. Only overexpression of the dehydratase operon *hadABC* or its dimer *hadBC* resulted in highly TAC-resistant *M. tuberculosis* strains (MIC > 0.2 mm) [170]. Curiously, the level of TAC resistance was much lower when the dimer *hadAB* was overexpressed in *M. tuberculosis* (MIC 10 μm). This was puzzling since HadAB is the target of TAC. Moreover, sequencing of spontaneous *M. tuberculosis* TAC-resistant mutants revealed mutations in HadA (Cys61Ser, Cys61Gly) but also in HadC (Val85Phe, Thr123Ala, Lys157Arg, Ala151Val) [170,174].

Grzegorzewicz and colleagues set up to unmask the role of *hadC* in TAC resistance [175]. The authors found that mutations in *hadC* protected *M. tuberculosis* from TAC and compensated for HadAB inhibition by TAC. The authors proposed that *hadC* mutations prevented TAC from reaching the active site of HadAB [175].

TAC resistance is also mediated by mutations in *mmaA4* (*Rv0642c*), a gene encoding a methoxy mycolic acid synthase required for the synthesis of keto- and methoxy-mycolic acids [170,176]. The methyltransferases involved in the modifications of the meromycolic acids such as MmaA4 interact with the proteins of the FAS-II system including the dehydratase heterodimers HadAB and HadBC [31]. Mutations in *mmaA4* might, therefore, modify the conformation of HadAB preventing the binding of TAC to HadA and causing TAC resistance [175].

*Area of investigation.*In the late 1940s, the discovery of the antiTB activity of the thiosemicarbazide TAC propelled pharmaceutical companies into an extensive search for TAC analogs with antiTB properties. This led to the discovery of the most important antimycobacterial drug INH (see above). This family of compounds might still reveal interesting molecules with pharmaceutical properties. Recently, new TAC analogs were synthesized with promising activities against *M. tuberculosis* [174].

## 4. New Generation of Cell Wall Inhibitors

### 4.1. Target: Mycolic Acids

#### 4.1.1. SQ109

*Discovery.* A collaboration between a pharmaceutical laboratory (Sequella) and an academic (Laboratory of Host Defenses NIAID/NIH) laboratory led to the synthesis and screening of a chemical library composed of 63,238 molecules based on the ethylenediamine core of EMB [177]. EMB was targeted for its good antimycobacterial activity in TB patients and lack of previous SAR studies. The goal was to create a more potent yet less toxic EMB analog. The screening selected 170 compounds that had an MIC of less than 6 mg/L against *M. tuberculosis* (discarding inactive or compounds that could not cross the cell wall) and had an inhibitory effect on the cell wall (determined by monitoring upregulation of the *iniBAC* operon promoter, a phenotype observed in cell wall inhibitors [178]). Based on the molecular structures of these 170 compounds, a new chemical library composed of 30,000 molecules was synthesized and retested against *M. tuberculosis* (in vitro and in macrophages). Eleven compounds passed those screens and were then assessed in mice [179]. The compound with the best antimycobacterial activity, pharmacological and toxicity data was SQ109.

*Activity.* SQ109 is a lipophilic, oral drug with poor water solubility (1.7 mg/L). SQ109 is bactericidal against drug-susceptible, MDR and XDR *M. tuberculosis* strains, with a MIC ranging from 0.4 to 3 μm. In vitro, SQ109 synergizes with the first-line TB drugs INH and RIF as well as with second-line TB drugs such as cycloserine, moxifloxacin, amikacin and bedaquiline (BDQ) [180]. In *M. tuberculosis*-infected mice, treatment with SQ109 for 28 days resulted in a dose-dependent reduction in lung and spleen burdens [181]. SQ109 was not as effective as INH in eliminating *M. tuberculosis* with organ burdens between 1 and 2.5 $\log_{10}$ higher in the SQ109 treatment compared to INH treatment [181]. However, SQ109 peak concentration in the lung and spleen was higher than SQ109 MIC and 45-fold higher than in plasma [181].

*Mode of action.* Surprisingly, this double SAR based on EMB resulted in a drug, SQ109, with a different mode of action than EMB. EMB-resistant *M. tuberculosis* strains were fully susceptible to SQ109 [179]. Transcriptional response of SQ109-treated *M. tuberculosis,* while consistent with other cell wall inhibitors, did not match EMB's response [182]. Lipid analysis of SQ109-treated *M. tuberculosis* revealed an inhibition of mycolic acid attachment to the arabinogalactan, a depletion of TDM and an accumulation of TMM [183]. To decipher the mechanism behind this result, isolation of spontaneous SQ109-resistant mutants was attempted without success (the mutation rate for SQ109 is exceptionally low ($10^{-11}$)) [183]. Instead, an analog of SQ109 was used to isolate spontaneous resistant mutants. These

mutants were co-resistant to SQ109 and carried a mutation in the mycolic acid flippase encoded by *mmpl3* (*Rv0206c*), involved in the translocation of TMM across the plasma membrane [184]. Analyses carried out in *M. smegmatis* revealed that SQ109 binds to Mmpl3 at the proton transportation site inhibiting the proton motive force for substrate translocation [185].

*Clinical trial.* In a 14 day study where SQ109 was given alone or in combination with RIF, SQ109 was shown to be safe in TB patients, with only gastrointestinal issues being disclosed [186]. The SQ109 plasma concentration was lower than the MIC for the drug and, in contrast to the in vitro results, the combination of SQ109 and RIF led to a decrease in SQ109 availability [186]. In mice, SQ109 had been shown to be metabolized by the cytochrome P450 isoenzymes $CYP_2D_6$ and $CYP_2C_{19}$ [181]. Since RIF induces the expression of $CYP_2C$ enzymes in humans [187], RIF might lower the effective dose of SQ109. On the other hand, the presence of SQ109 had no effect on the plasma concentration of RIF. The result of this 14 day trial was that SQ109 alone had no bactericidal effect, and there was no synergy or additive effect with the combination RIF and SQ109 [186]. In a phase 2 trial (NCT01785186) performed in newly diagnosed pulmonary TB patients, treatment with SQ109 in combination with RIF/INH/PZA failed to improve culture conversion compared to RIF/INH/PZA [188]. A phase 2b clinical trial in Russia on MDR TB patients revealed that SQ109, combined with standard drug therapy, was safe, well tolerated and effective (80% of sputum negative patients after 24 weeks of treatment with SQ109 compared to 61% without SQ109) [189].

*Area of investigation.* With the success of SQ109 and the essentiality of *mmpl3* in *M. tuberculosis*, novel inhibitors of Mmpl3 have been actively sought. Inhibition of Mmpl3 is not specific to the chemical structure of SQ109 and numerous chemical scaffolds are being identified as Mmpl3 inhibitors [185,190,191].

### 4.1.2. Pretomanid (PA-824)

*Discovery.* Another challenge in TB drug discovery is to discover new molecules that will eliminate dormant or persistent *M. tuberculosis,* a subpopulation of bacteria that are genetically drug-susceptible but 'resistant' (or tolerant) to drug treatment. One of the most used methods to study persistent *M. tuberculosis* is the Wayne model where *M. tuberculosis* is subjected to gradual oxygen depletion to allow for a slow entry into anaerobiosis. To find potential inhibitors of dormant *M. tuberculosis,* nitroimidazoles were tested, as they inhibit bacterial anaerobes. Metronidazole, an antibacterial and antiprotozoal drug with activity against Gram-positive and Gram-negative anaerobes [192] showed little, if any, activity against *M. tuberculosis* in mice [193]. However, a bicyclic nitroimidazole (CGI 17341) was reported to have potent in vitro and in vivo antimycobacterial activity. CGI 17341 showed no cross-resistance with other TB drugs but was highly mutagenic [194]. A series of more than 300 analogs was then synthesized to solve the CGI 17341 mutagenicity issue [195]. This led to the discovery of the nitroimidazole PA-824 now called pretomanid (PMD).

*Activity.* PMD is an oral drug, poorly soluble in water (0.012 g/L). PMD is active against replicating and non-replicating (anaerobic), drug-susceptible and drug-resistant (MIC ranging from 0.08 to 0.7 μm) *M. tuberculosis* strains. In a mouse model of *M. tuberculosis* infection, PMD given orally for 10 days performed as well as INH in lowering organ bacterial burdens. Increasing the concentration of PMD (from 25 mg/kg to 100 mg/kg) led to a significant reduction in organ burden compared to INH [195]. Similar results were obtained in *M. tuberculosis*-infected guinea pigs treated with PMD. The toxic threshold in mice was 1 g/kg for a single dose of PMD and 0.5 g/kg for a daily dose given for 28 days [195]. In a drug combination experiment, the lung burden in mice infected with the drug-susceptible *M. tuberculosis* H37Rv strain was 2.5 $\log_{10}$ lower in mice receiving PMD/BDQ/PZA than in mice treated with standard regimen (INH/RIF/EMB/PZA) after one month of treatment. None of the mice treated for 2 months with PMD/BDQ/PZA relapsed (mice were kept for an additional three months at the end of the treatment to assess relapse). The addition of moxifloxacin (MXF) to this combination did not improve the outcome during the first month of treatment but did decrease the number of mice relapsing after 1.5 month treatment [196]. In a subsequent experiment,

*M. tuberculosis*-infected mice were treated with BDQ/MXF/PZA +/− PMD [197]. The addition of PMD to the BDQ/MXF/PZA treatment decreased the lung burden an extra 1 $\log_{10}$ after one month of treatment compared to BDQ/MXF/PZA alone. In both groups, mice were relapse free after two months of treatment [197]. In addition, treatment of *M. tuberculosis*-infected mice with BDQ/linezolid (LZD) +/− PMD revealed that the addition of PMD had a significant impact on the lung burden. PMD in combination with BDQ/LZD resulted in a 7.2 $\log_{10}$ CFU *M. tuberculosis* killing in the lungs and relapse-free mice after two and three months of treatment, respectively. In contrast, the BDQ/LZD treatment resulted in a 5.2 log reduction in CFU in the lungs, but more than 90% of the mice relapsed even after 4 months of treatment [197].

*Mode of action.* Like many other TB drugs, PMD is a prodrug. PMD is activated by a deazaflavin (F420)-dependent nitroreductase (Ddn, Rv3547) to generate a des nitro metabolite releasing nitric oxide (NO) [198]. The presence of the des nitro metabolite and rate of NO release were linked specifically to the anaerobic killing activity of PMD. PMD treatment of *M. tuberculosis* also causes a decrease in ketomycolates production and an accumulation of hydroxymycolates leading to the hypothesis that PMD inhibits an enzyme or affects a cofactor responsible for the oxidation of hydroxymycolic acids to ketomycolic acids [195]. A transcriptional analysis of *M. tuberculosis* treated with PMD under replicating conditions revealed that PMD action had similarity with cell wall inhibitors as well as inhibitors of the respiratory chain [199]. This data confirmed the dual action of PMD as an inhibitor of a subfamily of mycolic acids as well as a NO generator leading to the inhibition of the respiratory chain under anaerobic conditions. This combination of a cell wall inhibitor and NO generator makes PMD a unique antimycobacterial drug, effective against active and dormant TB that can shorten the duration of chemotherapy.

*Resistance.* The main mechanism of resistance to PMD is through mutations in its activator Ddn. In vitro selection of spontaneous PMD-resistant mutants in *M. tuberculosis* led to the isolation of 183 mutants [200]. The mutation rate was relatively high ranging from $10^{-5}$ to $10^{-7}$. Most of the mutations were in *ddn* (SNPs, base pair deletion or insertion, early stop codon). Mutations in *fbiABC* (*Rv3261, Rv3262, Rv1173*) encoding F420 biosynthesis proteins and *fgd1* (*Rv0407*) encoding a F420-dependent glucose-6-phosphate dehydrogenase were also identified, but no complementation experiments were performed to confirm the role of these mutations in PMD resistance.

*Clinical trial.* A phase 2a clinical trial (NCT01215851) evaluated the early bactericidal activity (EBA), safety and tolerability of PMD combined with BDQ, PZA and/or MXF in newly diagnosed drug-susceptible, smear-positive pulmonary TB patients [201]. These combinations were well tolerated and seemed safe. The combination PMD/PZA/MXF was more bactericidal than PMD/BDQ or PMD/PZA and as potent as the standard regimen (INH/RIF/EMB/PZA) [201]. In a phase 2b clinical trial (NCT02193776), the efficacy, safety and tolerability of the combination PMD/BDQ/PZA were assessed in newly diagnosed drug-susceptible, smear-positive pulmonary TB patients treated for 8 weeks [202]. TB patients converted more rapidly to sputum negativity with PMD/BDQ/PZA than the standard regimen. One arm of the study also looked at the combination PMD/BDQ/PZA/MXF for the treatment of newly diagnosed, MXF-sensitive, MDR, pulmonary TB patients. MDR TB patients converted to sputum negativity within 8 weeks of treatment with PMD/BDQ/PZA/MXF. A phase 2c trial (NCT03338621) is currently in progress to evaluate the efficacy, safety and tolerability of a 4 month PMD/BDQ/PZA/MXF treatment given to patients infected with drug-sensitive TB or a 6 month treatment given to drug-resistant TB patients. No results have been posted yet.

The Nix-TB trial (phase 3, NCT02333799) was set to test the efficacy, tolerability, safety and pharmacodynamics of the combination PMD/BDQ/LZD given for six months on 50 patients infected with XDR TB and 24 patients infected with treatment-intolerant or non-responsive MDR TB. 88% of XDR TB cases and 92% of MDR TB cases had favorable outcomes at the end of treatment (no clinical infection, culture negative 6 months post treatment). 23% of the patients had manageable adverse effects [203]. On August 14, 2019, the Food and Drug Administration (FDA) approved the use of PMD in combination with BDQ and LZD for the treatment of pulmonary TB only in the case of non-responsive MDR, XDR

and treatment-intolerant *M. tuberculosis* infections. This new regimen is all oral, short (6 months), more efficacious (sputum conversion in less than 6 weeks) and uses fewer drugs (3) compared to the treatment for highly resistant TB, which entails daily injections for 6 months followed by daily treatment with five drugs for 12 to 18 months (https://www.fda.gov/media/128001/download).

### 4.1.3. Delamanid (OPC-67683)

*Discovery.* Upon the discovery of the activity of CGI 17341 against *M. tuberculosis*, researchers at Otsuka Pharmaceutical Co., Japan, performed a SAR study on the compound to reduce its mutagenicity and increase its potency against *M. tuberculosis* [204]. The result was the nitroimidazole OPC-67683, now called delamanid.

*Activity.* Delamanid is an oral, non-mutagenic compound, poorly soluble in water (0.002 g/L). Delamanid is active in vitro (MIC 8–45 nM [205]) against drug-susceptible and MDR *M. tuberculosis* strains and highly efficacious in vivo [206]. Delamanid is also active against non-replicating *M. tuberculosis* [207]. Delamanid is only active against members of the MTB complex as well as some non-tuberculous mycobacteria and shows no activity against bacterial microflora [205]. In a mouse model of acute and chronic *M. tuberculosis* infections, delamanid performed better than PMD. After 3 weeks of treatment, in the acute model of infection, delamanid and PMD reduced the lung bacterial burden by 1 $\log_{10}$ and 0, respectively. In the chronic model of infection, delamanid and PMD reduced the lung bacterial burden by 3 and 2 $\log_{10}$, respectively [207]. Delamanid is included in the Group D2 (add-on agents) of second-line TB drugs to treat RIF-resistant and MDR TB patients.

*Mode of action.* Similar to PMD, delamanid is a prodrug activated by the deazaflavin-dependent nitroreductase Ddn to form a des nitro metabolite and release NO. Delamanid inhibits the biosynthesis of methoxy- and keto-mycolic acids but not of α-mycolic acids [206]. Deciphering the mechanism of action of delamanid by isolating spontaneous in vitro resistant mutants has led to the identification of mutations in genes involved only in delamanid activation (*ddn, fgd1, fbiABC*), not in its target [208]. Frequency of mutation was relatively high ($10^{-5}$ to $10^{-6}$). Complementation restored drug susceptibility except for one *fbiB* mutant that required complementation with a plasmid containing both *fbiB* and *fbiA*. Two delamanid-resistant clinical isolates were also analyzed in that study and shown to have mutations in *ddn* (L107P and a 59–101 bp deletion). The target of delamanid is yet to be discovered, as no mutant in pathways independent to delamanid's mode of activation has been isolated. The question remains as to whether delamanid and PMD target mycolic acid biosynthesis or whether the inhibition of the biosynthesis of these specific mycolic acids observed during delamanid or PMD treatment of *M. tuberculosis* is only a consequence of the inhibition of these compounds' target(s).

*Clinical trial.* As of 2019, delamanid is one of three new TB drugs in phase 3 clinical development along with PMD and BDQ (www.newtbdrugs.org). In a clinical trial in MDR TB patients, delamanid was added for 2 months to an optimized background treatment regimen (OBR), resulting in 45% sputum culture conversion compared to 30% sputum culture conversion for the patients getting OBR and placebo [209]. In this study, the group of patients receiving higher doses of delamanid (200 mg, twice a day) had a higher incidence of palpitation and prolonged QT intervals than the groups that received a lower concentration (100 mg, twice a day) of delamanid or placebo. In a subsequent six-month delamanid trial, a lower mortality rate was observed in TB patients that received delamanid for 6 months rather than the previous 2 months trial, but no significant difference in successful treatment outcome was recorded between the 2 month and 6 month delamanid trial [210,211]. A phase 3 clinical trial on MDR TB patients tested the addition of delamanid for the first six months of the 24 month OBR regimen. There was no statistical difference in time to sputum culture conversion or rate of adverse events whether the patients received delamanid or not with their OBR treatment [212]. Overall, delamanid is a well-tolerated drug with good safety data. In 2014, delamanid was approved for the treatment of pulmonary MDR TB in adults in Europe, Japan and Korea.

Delamanid, PMD and SQ109 are the only drugs in advanced clinical development so far targeting mycolic acid biosynthesis or mycolic acid incorporation into the cell wall. Academic and pharmaceutical

laboratories are developing new inhibitors of enzymes involved in mycolic acid biosynthesis. Inhibitors of KasA [213], InhA [83,90–92], Pks13 [214–217] and Mmpl3 [185,190,191,218,219] are being tested.

*4.2. Target: Arabinogalactan/LAM*

4.2.1. CPZEN-45

*Discovery.* Caprazamycin B, a liponucleoside isolated from *Streptomyces,* is a non-toxic antibiotic with good activity in vitro (MIC 3–11 μm) against drug-susceptible and MDR *M. tuberculosis* strains but insoluble in water [220]. To increase its hydrophilicity, SAR was performed and led to the nucleoside caprazene-45 (CPZEN-45) [221].

*Activity*. CPZEN-45 is a water-soluble drug (solubility ≈ 10 g/L) with an MIC of 2–5 μm and 10 μm against drug-susceptible and MDR *M. tuberculosis,* respectively. Unlike caprazamycin B, which is active against several Gram-positive bacteria, CPZEN-45 has no activity against *S. aureus, Streptococcus pneumonia* or *Enterococcus faecalis* [222]. CPZEN-45 is specific to slow-growing pathogens. In mice intravenously infected with the drug-susceptible *M. tuberculosis* H37Rv strain, a 30-day subcutaneous treatment with CPZEN-45 was as effective in reducing lung burden as INH and better than RIF alone [223]. Furthermore, the combination INH/RIF/CPZEN-45 resulted in at least a 1 $\log_{10}$ better killing of *M. tuberculosis* than the combination INH/RIF. In a subsequent experiment, mice were intravenously infected with an XDR *M. tuberculosis* strain and treated with CPZEN-45 at doses ranging from 6.3 to 200 mg/kg. After 30 days, the mice treated with the highest concentration of CPZEN-45 had a 1.5 $\log_{10}$ better reduction in CFUs in the lungs than the mice receiving the lowest concentration [223].

*Mode of action.* CPZEN-45 is an inhibitor of *M. tuberculosis* WecA (Rv1302) with an $IC_{50}$ of 7 nM [222]. WecA is involved in the first step of the arabinogalactan biosynthesis forming the anchor point between peptidoglycan and arabinogalactan. Transcriptionally silencing *wecA* is bactericidal in *M. tuberculosis* in vitro and bacteriostatic ex vivo, validating WecA as candidate for drug development [224].

*Clinical trial.* CPZEN-45 is in the early stage of clinical development.

4.2.2. BTZ043

*Discovery.* A sulfur-based chemical library was tested for antibacterial and antifungal activities. A class of compounds, the nitrobenzothiazinones, was found to have specific activity against mycobacteria. The most promising hit BTZ038 was a racemic molecule and synthesis of its *S* enantiomer gave BTZ043 [225].

*Activity.* BTZ043 is lipophilic, bactericidal, and is active against drug-susceptible, MDR and XDR *M. tuberculosis* strains, with an MIC ranging from 2 to 70 nM. In contrast to the novel TB drugs targeting the mycolic acids, BTZ043 was less effective in non-replicating conditions, suggesting that BTZ043 would have to be used in combination with other drugs to be effective against TB [225]. BTZ043 was tested with first-, second-line and in-development TB drugs and showed no antagonistic effects. BTZ043 was additive with most of the drugs tested and synergistic with BDQ [226]. The compound is as effective in a mouse model of *M. tuberculosis* infection as INH. BTZ043 was well tolerated in rats, had low interaction with the CYP450 enzymes and showed no mutagenic or genotoxic properties [227].

*Mode of action.* BTZ043 is a prodrug activated by DprE1 (Rv3790). DprE1 reduces the nitro group in BTZ043, yielding a nitroso metabolite. This metabolite reacts with the thiol group of a cysteine (Cys387) residue in the substrate-binding site of DprE1 to form a covalent bond that irreversibly inhibits DprE1, classifying BTZ043 as a suicide inhibitor [228]. DprE1 catalyzes the first step in the epimerisation of decaprenylphosphoryl ribose (DPR) to decaprenylphosphoryl arabinose (DPA), the arabinose donor for the synthesis of arabinogalactan and LAM. Treatment of *M. tuberculosis* with BTZ043, therefore, results in inhibition of DPA formation and ultimately inhibition of both arabinogalactan and LAM [225].

*Resistance.* Mutation at the Cys387 position of *dprE1* is the main mechanism of resistance to BTZ043. Frequency of mutation is low ($<10^{-8}$); however, the resulting mutants *dprE1* Cys387Ser and Cys387Gly are highly resistant to BTZ043 (250-10,000-fold) [225].

*Clinical trial.* As of December 2019, BTZ043 is in Phase 1b/2a clinical trial to assess safety, tolerability, interaction with first-line TB drugs and early bactericidal activity in newly diagnosed patients infected with uncomplicated, smear-positive, drug-susceptible TB.

### 4.2.3. PBTZ169

*Discovery.* SAR on BTZ043 to improve its pharmacologic properties yielded PBTZ169 (now called macozinone), a piperazine derivative. Compared to BTZ043, PBTZ169 has the advantage of having no chiral center which facilitates its chemical synthesis.

*Activity.* PBTZ169 is a very potent bactericidal benzothiazinone with an MIC of 0.6 nM against *M. tuberculosis*. PBTZ169 has better potency, pharmacodynamics and is 10 times less cytotoxic than BTZ043. PBTZ169 has poor solubility in water (0.9 g/L). PBTZ169 is synergistic with BDQ and clofazimine (CFZ) but not with other new TB drugs such as delamanid, linezolid, meropenem or sutezolid [229]. Interestingly, this synergistic phenotype was also observed in non-replicating conditions where PBTZ169 has no activity [230]. In a mouse model of chronic *M. tuberculosis* infection, the reduction in lung and spleen burden was similar between INH and PBTZ169 after 4 weeks of treatment [231]. Furthermore, in a mouse model of chronic *M. tuberculosis* infection, the lung burden was reduced by 2, 4 and 4.6 $\log_{10}$ after 28 days of treatment with PBTZ169, CFZ and the combination PBTZ169/CFZ, respectively [230]. In a similar experiment, the combination PBTZ169/BDQ/PZA was also shown to be more efficient in reducing lung and spleen burden of chronically *M. tuberculosis*-infected mice than the standard INH/RIF/PZA treatment [231].

*Mode of action.* PBTZ169 mode of action is similar to BTZ043. PBTZ169 is a prodrug, activated by DprE1 to yield a nitroso metabolite that covalently binds to the Cys387 residue of DprE1.

*Clinical trial.* A phase I study (NCT03423030) in healthy male subjects receiving increasing doses of PBTZ169 showed that PBTZ169 was well tolerated and safe. No result has been posted for a phase 2a EBA study (NCT03334734) where PBTZ169 was given as a single dose to TB patients.

### 4.2.4. OPC167832

*Activity.* OPC167832 was developed by Otsuka pharmaceuticals. OPC167832 is a lipophilic, bactericidal compound, active against drug-susceptible, MDR and XDR *M. tuberculosis* strains, with a MIC ranging from 0.5 to 5 nM. OPC167832 is active in macrophage and in mouse model of *M. tuberculosis* infection at a dose of 1.25 mg/kg. OPC167832 is not antagonistic with other TB drugs. The combination of OPC167832 with delamanid and other TB drugs showed better efficacy than standard TB treatment in a mouse model of chronic drug-susceptible and MDR *M. tuberculosis* infections [232].

*Mode of action.* OPC167832 is a DprE1 inhibitor. OPC167832 does not contain a nitro group, and so it inhibits DprE1 via other interactions than a covalent bond with DprE1 Cys387 (see below).

*Resistance.* Spontaneous OPC167832-resistant mutants isolated at 16 × MIC occurred at a frequency of $2.6 \times 10^{-9}$ to $1.5 \times 10^{-7}$ in *M. tuberculosis* H37Rv. Mutations in *dprE1* and *Rv0678* encoding a transcriptional regulator of the efflux pumps MmpS5 and Mmpl5 [233] drive resistance to OPC167832.

*Clinical trial.* A phase 1/2 clinical trial (NCT03678688) is recruiting to test the safety, tolerability and pharmacokinetics of OPC167832 given at increasing concentrations to patients infected with uncomplicated, smear-positive, drug-susceptible TB (phase I). A phase II study will compare delamanid and OPC167832 vs. delamanid only or INH/RIF/EMB/PZA to demonstrate that this new regimen, which uses only oral drugs, is safer and can shorten TB treatment [232].

### 4.2.5. TBA-7371

***Discovery.*** TBA-7371 was developed in collaboration between Astra Zeneca and the TB Global Alliance. TBA-7371 is a derivative of 1,4-azaindoles developed through scaffold morphing of an imidazopyridine compound with good MIC but low minimum bactericidal activity (MBC) against *M. tuberculosis* [234]. SAR on the hit azaindole compound led to TBA-7371.

***Activity.*** TBA-7371 is bactericidal against *M. tuberculosis*, with a MIC ranging from 0.78 to 3.12 μm against drug-susceptible and drug-resistant clinical isolates of *M. tuberculosis* [235]. Solubility of TBA-7371 is 170 μm. TBA-7371 does not exhibit toxicity in THP1 cell line up to concentration of 0.1 mM [235].

***Mode of action.*** TBA-7371 inhibits DprE1 with an $IC_{50}$ of 10–30 nm but, unlike BTZ043 and PBTZ169, TBA-7371 is a non-covalent inhibitor of DprE1 [234].

***Resistance.*** Resistance to TBA-7371 was observed in strains overexpressing *M. tuberculosis dprE1* or carrying the mutation Y314H in *dprE1*, but not in strains with a mutated Cys387 in *dprE1* [234].

***Clinical trial.*** TBA-7371 is in phase I clinical trial (NCT03199339) for safety, tolerability and pharmacokinetic studies in healthy adults. No results have been posted.

DprE1 is a successful drug target. Since the discovery of BTZ043 and its target DprE1 [225], numerous groups have uncovered novel molecules that target DprE1. DprE1 inhibitors can be divided into two groups depending on whether or not they bind covalently to DprE1. The presence of the nitro group is a requirement for the covalent binding to the Cys387 amino acid of DprE1. Among the covalent inhibitors are benzothiazinethione [236], dinitrobenzamides [237], nitroquinoxalines [238] and nitroimidazoles [239]. Several non-covalent DprE1 inhibitors have been identified with various chemical structures: azaindoles [234], benzothiazoles [240], pyrazolopyridones [241], aminoquinolone piperidine amides [242], carboxyquinoxalines [243], pyrrole-benzothiazinones [244], pyridobenzimidazole [245], sulfonylpiperazin-benzothiazinones [246] and piperidinopyrimidines [247]. These compounds inhibit DprE1 through electrostatic or hydrophobic interactions with different sites of DprE1. Most of these compounds have low MICs (in the nM range), are active against MDR and XDR TB as well as in a mouse model of *M. tuberculosis* infection. Considering the promiscuity of the DprE1 target, better DprE1 inhibitors might still be discovered to improve TB treatment.

### *4.3. Target: Peptidoglycan*

### 4.3.1. β-Lactams and Clavulanic Acid

***Discovery.*** The first class of synthetic drugs that successfully treated bacterial infections were the sulfa drugs in the early 1930s [163]; however, by the early 1940s, penicillin had quickly eclipsed the sulfa drugs. Penicillin, a β-lactam compound, was active against Gram-positive and Gram-negative, yet penicillin, like the sulfa drugs, had no activity against *M. tuberculosis*. For penicillin, the lack of activity was due to the presence in *M. tuberculosis* of a potent β-lactamase BlaC. The innate resistance of *M. tuberculosis* to β-lactams decreased when blaC was deleted from *M. tuberculosis* or when the β-lactamase irreversible inhibitor clavulanic acid [248] was used in combination with β-lactams [249]. Hugonnet and colleagues used this knowledge to validate the combination of meropenem, a carbapenem with poor affinity for BlaC, and clavulanic acid as an efficacious *M. tuberculosis* inhibitor [250].

***Activity.*** The combination meropenem plus clavulanate is part of the group D3 (add-on agents) of second-line TB drugs. Meropenem is a potent inhibitor of drug-susceptible and XDR *M. tuberculosis* strains (MIC range from 0.6 μm to 3.3 μm) when combined with clavulanate [250]. Interestingly, this combination also inhibits non-replicating *M. tuberculosis* [250]. In a chronic model of *M. tuberculosis* infection in mice, meropenem had a modest activity in reducing lung burdens [251]. Surprisingly, the addition of clavulanate to the meropenem treatment did not increase meropenem activity in mice. In mice intravenously infected with *M. tuberculosis*, the lung burden had increased by almost a 1 $\log_{10}$ after 4 weeks of treatment with meropenem and clavulanate showing no inhibitory activity of this

combination in vivo [252]. However, studies in mice are problematic since the mice dehydropeptidase (DHP) rapidly cleaves the β-lactam ring.

Meropenem is administered via intravenous injection and new regimens are steering away from injectables. Therefore, Dhar and colleagues tested faropenem, an *oral* β-lactam, and demonstrated activity against *M. tuberculosis* (MIC 4.6 μm) without clavulanate addition (meropenem MIC is 8-fold higher without clavulanate) [253]. Faropenem is also more soluble in water than meropenem (solubility 14.7 g/L vs. 8 g/L for meropenem). Using the dehydropeptidase inhibitor, probenecid, Dhar and colleagues demonstrated that *M. tuberculosis*-infected mice had a small but significant reduction in lung burden after 9 days of treatment with a combination faropenem/clavulanate/probenecid. In a study using DHP-I knockout mice [254], mice were infected intratracheally with *M. tuberculosis* H37Rv and treated for 8 consecutive days, 10 to 12 days post infection. Meropenem was given subcutaneously three times a day in combination with amoxicillin and clavulanate. Another group of mice received faropenem orally with amoxicillin and clavulanate. A control group was treated with moxifloxacin. Treatment also consisted of a dose of probenecid prior to drug administration. While the lung burden was 7.7 $\log_{10}$ CFU/mouse at the beginning of the treatment, the lung burdens at the end of the treatment were 7.8 and 7.5 $\log_{10}$ CFU/mouse for the meropenem and faropenem groups, respectively. Only the mice receiving moxifloxacin had a 2-$\log_{10}$ reduction in lung burden [254].

Experiments performed in other animal models (rabbits and monkeys) revealed that intravenous injections of meropenem with or without clavulanate and dehydropeptidase inhibitor led to serious adverse effects (diarrhea, weight loss and death) [251].

*Mode of action.* Meropenem binds to and inhibits the L,D-transpeptidase $Ldt_{Mt1}$ (Rv0116c), which catalyzes the 3→3 cross-linkage in the peptidoglycan [255,256]. Faropenem is 14-fold more potent in inactivating $Ldt_{Mt1}$ than meropenem, but this did not correlate with increasing killing against intracellular *M. tuberculosis* [253]. Dhar and colleagues demonstrated that the addition of faropenem to *M. tuberculosis* quickly arrested growth, which did not translate into an immediate lysis of the cells. This is opposite to the current understanding that β-lactams induce cell lysis by inhibiting peptidoglycan cross-linking while cells are still actively dividing. Dhar and colleagues concluded that the mechanism of *M. tuberculosis* killing by β-lactam should be revisited [253].

*Clinical trial.* A clinical study on MDR and XDR TB patients receiving meropenem/clavulanate (intravenous injections three times a day for an average of 85 days) in combination with an MDR/XDR drug regimen showed that the addition of meropenem/clavulanate did not improve: (1) sputum smear or culture conversion; (2) treatment outcome; or (3) treatment success [257]. A phase 2 clinical trial evaluated the early bactericidal activity, safety and tolerability of intravenously administered meropenem with amoxicillin and clavulanate versus orally-administered faropenem with amoxicillin and clavulanate (NCT02349841). Results have not been posted yet.

### 4.3.2. Sanfetrinem

*Activity.* Sanfetrinem is a tricyclic β-lactam developed by GlaxoSmithKline in the 1990s. Sanfetrinem has broad-spectrum activity against Gram-negative and Gram-positive bacteria. Sanfetrinem is also very stable to β-lactamases and to human renal dehydropeptidase (DHP) [258]. The MICs for sanfetrinem against *M. tuberculosis* H37Rv ranged from 1.25 to 5 μm with clavulanate (2.5 to 7.5 μm without clavulanate). In a checkerboard assay, the combination of sanfetrinem with RIF or amoxicillin was synergistic, but not with delamanid or ethambutol. Sanfetrinem cilexetil is an oral prodrug ester of sanfetrinem and has an MIC ranging from 5 to 20 μm with clavulanate (7.5 to 20 μm without clavulanate) against *M. tuberculosis*. The compounds performed better against intracellular *M. tuberculosis*. In THP1 monocytes, the MIC for sanfetrinem and sanfetrinem cilexetil ranged from 2.1 to 7.7 μm without clavulanate (6.1 to 8.8 μm with clavulanate) and 3.4 to 7.0 μm without clavulanate (4.6 to 5.3 μm with clavulanate), respectively. In a mouse model of acute *M. tuberculosis* infection, DHP-1 KO mice were infected intratracheally with *M. tuberculosis* H37Rv. Treatment started 9 days post infection, twice a day, for 5 days. Sanfetrinem was given subcutaneously, while sanfetrinem cilexetil

and clavulanate were given orally. Growth arrest but not killing of *M. tuberculosis* was observed during this five-day treatment [259].

*Clinical trial.* A phase 2a clinical study is projected to start the first quarter of 2021 [260].

## 5. Conclusions

In 1882, when Robert Koch discovered the causative agent of TB, there was no valid treatment for the disease. In the past 70 years, with the introduction of multidrug chemotherapy, the cure rate for drug-susceptible *M. tuberculosis* infection has reached up to 95%. The treatment is effective but very long, with drugs that may not be well tolerated. New chemotherapy regimens are needed with drugs that are less toxic and more efficacious, so that a shorter treatment can be achieved. The pursuit of mycobacterial cell wall inhibitors has offered several hits with novel modes of action and remarkable potency that have reached clinical trials. Those pioneering compounds can lead the way to more valuable therapeutics. Numerous new compounds are being studied that either derive from the known hits or target other biosynthetic pathways of the cell wall [261,262]. In the past 20 years, *M. tuberculosis* has been unraveling many of its secrets: its niches in the host, its metabolism and its way of fighting drugs. New tools were developed to study *M. tuberculosis* and to search for new TB drugs more efficiently combining target and whole-cell screenings. We should be closer to a new TB regimen for a shorter treatment of drug-susceptible TB and a more successful treatment of MDR and XDR TB. The collaboration between academic and pharmaceutical laboratories and the involvement of funding foundations have opened the doors to the clinical testing of new drugs and new regimens [263].

In Selman Waksman's book "The Conquest of Tuberculosis" [264], a quote from Georges Canetti's speech at the 1961 Sixteenth International Conference on Tuberculosis, as reported by the chairman, stated: "We are not concerned with eradication in the absolutely literal sense of the word because this is something that many of us believe to be biologically impossible. What we are talking about, however, is tuberculosis 'eradication' in the sense of reducing the problem to the point where the disease is a scientific curiosity. This is a biologic possibility". With better diagnostic tools, a more efficient vaccine and sterilizing therapeutics, this might become a biologic reality.

A summary of the cell wall inhibitors discussed herein is presented in Table 1 with their chemical structures and targets.

**Table 1.** TB drugs targeting the mycobacterial cell wall.

| Name | Structure | Cell Wall Component Inhibited | Prodrug/Activator | Target |
|---|---|---|---|---|
| Isoniazid |  | Mycolic acids | + KatG | InhA |
| Ethionamide Prothionamide |  | Mycolic acids | + EthA, MymA, Rv0565c | InhA |
| Thiacetazone |  | Mycolic acids | + EthA | HadAB |
| Pretonamid |  | Keto-mycolic acids | + Ddn | ? |

**Table 1.** *Cont.*

| Name | Structure | Cell Wall Component Inhibited | Prodrug/Activator | Target |
|---|---|---|---|---|
| Delamanid |  | Methoxy- and keto-mycolic acids | +<br>Ddn | ? |
| SQ109 |  | Mycolic acid transport | - | Mmpl3 |
| CPZEN45 |  | Arabinogalactan | - | WecA |
| Ethambutol |  | Arabinogalactan LAM | - | EmbCAB |
| BTZ043 |  | Arabinogalactan LAM | +<br>DprE1 | DprE1 |
| PBTZ169 |  | Arabinogalactan LAM | +<br>DprE1 | DprE1 |
| OPC167832 |  | Arabinogalactan LAM | - | DprE1 |
| TBA7371 |  | Arabinogalactan LAM | - | DprE1 |
| Cycloserine |  | Peptidoglycan | - | DdlA |

**Table 1.** *Cont.*

| Name | Structure | Cell Wall Component Inhibited | Prodrug/Activator | Target |
|---|---|---|---|---|
| Meropenem/ clavulanate | | Peptidoglycan | - | LdtM1 |
| sanfetrinem | | Peptidoglycan | - | |

**Funding:** CV acknowledges support through grant AI21670 from the US National Institutes of Health.

**Acknowledgments:** I thank Lawrence Leung, Claire Mulholland and Saranathan Rajagopalan for their critical reading, comments and feedback on this manuscript.

**Conflicts of Interest:** The author declares no conflict of interest.

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
