# Peer review of "Mycobacterial Cell Wall: A Source of Successful Targets for Old and New Drugs"

_applsci, doi:10.3390/app10072278_

Round 1

Reviewer 1 Report

The manuscript is extremely interesting, very well and clearly written.

The title is informative, clearly stated aim tally with the content of the paper.

The abstract created an expectation that a balanced, realistic reflection on the potential of drugs in TB studies would be discussed in the review. The effect is achieved in the body of the manuscript.

All references were cited in the text correctly.

Apart from some punctuation mistakes, the scientific level of the review is quite high and I have no doubts to suggest the manuscript for publication. 

Author Response

I thank the reviewer for the positive comments.

The inconstancies in the punctuation were corrected.

Reviewer 2 Report

The present review manuscript provides detailed insights into antibiotic compounds that act on the cell wall of M. tuberculosis and that are current in clinical use or under clinical investigation.

  • The historic context throughout the review is very interesting and increases the quality and readability of the review.
  • The review is very well written and represents a comprehensive database on cell wall-active anti-Tb drugs.
  • I was wondering if there went something wrong with the formatting of the references 3, 4, and 6 in the references section, which read “No authors available”
  • Page 4: Numbers of patients suffering from MDR/XDR Tb should be updated according to the Global Tuberculosis Report 2019 by WHO showing figures for 2018
  • I could imagine that the review may benefit from a summarizing scheme that depicts where the described antibiotics act on within the cell wall.
  • Overall, the review seems to be complete and thorough.

Author Response

I thank the reviewer for the positive comments.

The references 3, 4 and 6 do not have authors listed. I agree with the reviewer that the mention “No authors available” may be inappropriate so I modified the way these references are cited.

Page 4: The numbers of MDR cases have been updated according to the WHO report from 2019.

The sites of inhibition of the different TB drugs described in this manuscript have been added to Figures 1 and 2. Table 1 summarizes the mode of action of these drugs and has been moved to the end of the manuscript.